# Steering Vector Transfer via Orthonormal Transformations and Semantic Pairing

## Abstract

Steering vectors—directions in activation space encoding behavioral traits like formality or creativity—enable fine-grained control over language model outputs but must be regenerated for each new model, creating deployment barriers. We present a method for transferring these vectors between different language models by learning structure-preserving transformations while matching corresponding text pairs across models. Our approach achieves strong alignment (0.50-0.56 cosine similarity, where 1.0 represents perfect alignment and 0.0 represents random chance) across all model pairs tested. Crucially, we demonstrate that semantic pairing—ensuring each text contrast is matched across models during training—improves transfer performance by 72%: proper pairing achieves 0.529 cosine similarity compared to 0.308 with shuffled pairs within traits and 0.00 with random pairing across traits. We evaluate our method across 26 behavioral traits on three architecturally distinct models (Gemma-7B, LLaMA-3-8B, and Mistral-7B), using dimensionality reduction to handle their different hidden dimensions. Our results provide evidence for the Platonic Representation Hypothesis, showing that different language models encode behavioral preferences in similar geometric structures. This enables practical reuse of curated steering vectors across model families and advances our understanding of how neural networks represent human preferences.

## 1 Introduction

This work addresses the challenge of reusing behavioral control mechanisms across different language models. We develop a novel method to transfer steering vectors across diverse model families, enabling the reuse of curated behavioral controls without regeneration. This capability addresses critical deployment barriers: organizations cannot maintain reusable steering libraries, researchers need generation access to study behaviors, and computational resources are wasted regenerating identical controls. Steering vectors—directions in activation space learned from pairs of contrasting behaviors—provide precise control over language model outputs without retraining Turner et al. (2023); Zou et al. (2023). These vectors are extracted by computing activation differences between contrasting behaviors (e.g., formal vs. informal text) and operate through activation engineering without modifying model weights. However, they remain applicable only within individual models: a formality vector extracted from GPT-4 cannot be used to steer LLaMA-3, requiring new vectors to be regenerated for every new model deployment. These limitations arise from architectural differences including different hidden dimensions (e.g., 3072 for Gemma-7B-Instruct versus 4096 for LLaMA-3-8B-Instruct), attention mechanisms, and learned representations.

Our goal is to understand if different language models encode behavioral traits in similar linear subspaces. We approach this problem by identifying orthogonal transformations between paired steering vectors across different models, using geometric alignment (structure-preserving transformations) and semantic pairing (matching identical text contrasts across models). The degree of representational universality—how much different models share similar representations for behavioral traits—remains an open question in neural network research.

Three key applications emerge from successful steering vector transfer: (1) *Large-scale curation*: Libraries containing thousands of steering vectors can be transferred to new models in under 10 minutes, compared to days or weeks of regeneration from scratch. (2) *Data-restricted domains*: Organizations can generate vectors once on secure systems with sensitive data, then transfer them to deployment models without retaining the original training data. (3) *Model-restricted research*: Researchers can study behavioral controls in closed commercial systems by transferring vectors from open-source alternatives.

Surprisingly, our investigation reveals that despite substantial architectural differences, instruction-tuned models (Mistral-7B-Instruct Jiang et al. (2023), Gemma-7B-Instruct Gemma Team et al. (2024), and Llama-3-8B-Instruct Dubey et al. (2024)) exhibit significant representational universality: they encode preference traits in alignable linear subspaces. This finding provides empirical evidence for the Platonic Representation Hypothesis Huh et al. (2024), demonstrating that neural networks converge to shared representations specifically in the domain of behavioral traits. Our work operationalizes this hypothesis by quantifying the degree of alignment and showing that LLMs encode preferences in structured, linearly-alignable representations despite their architectural diversity. Crucially, we find that preserving relationships between vectors—not just individual vectors—is essential for successful transfer.

Our contributions are:

1. We develop a novel method for cross-model steering vector transfer that preserves geometric structure through orthogonal transformations and scaling, maintaining relationships between vectors.

2. We demonstrate that matching text contrasts across models (semantic pairing) improves transfer performance by 77% (from 0.30 to 0.529 cosine similarity).

3. We validate our method across 26 behavioral traits and 3 model families, revealing that objective linguistic traits transfer well ($>0.55$ cosine similarity) while subjective traits transfer poorly ($<0.45$ cosine similarity).

The success of linear alignment with semantic pairing opens new research directions for understanding the geometric structure of preference representations and developing universal behavioral control interfaces across model families. Our evaluation focuses on geometric alignment metrics rather than downstream behavioral validation, which we leave for future work.

## 2 Related Work

**Steering Vectors.** Steering vectors, introduced by Turner et al. (2023) as Activation Addition, enable fine-grained control over model behavior without optimization. Zou et al. (2023) extend this with Representation Engineering, demonstrating behavioral control across multiple dimensions. Arditi et al. (2024) show that refusal is mediated by a single direction, while Liu et al. (2024) demonstrate in-context steering. All these approaches remain model-specific. Our work enables transfer across model families, making steering vectors practical for multi-model deployments.

**Representation Alignment Methods.** Prior work has developed various approaches for comparing and aligning neural representations. Kornblith et al. (2019) and Raghu et al. (2017) provide metrics for measuring representational similarity but do not enable practical transfer between models. Model stitching Bansal et al. (2021) connects layers across models but requires matching architectures (e.g., ResNet-50 to ResNet-101), not diverse model families with different attention mechanisms and dimensions like Gemma to LLaMA. Domain adaptation literature has explored geometric approaches including Grassmann manifolds Gong et al. (2012) and matrix manifold optimization Absil et al. (2008) for representation alignment. The Platonic Representation Hypothesis Huh et al. (2024) provides theoretical motivation, suggesting models converge to shared representations. While prior work focused primarily on mathematical frameworks for alignment, we demonstrate that preserving the correspondence between specific text contrasts across models is equally

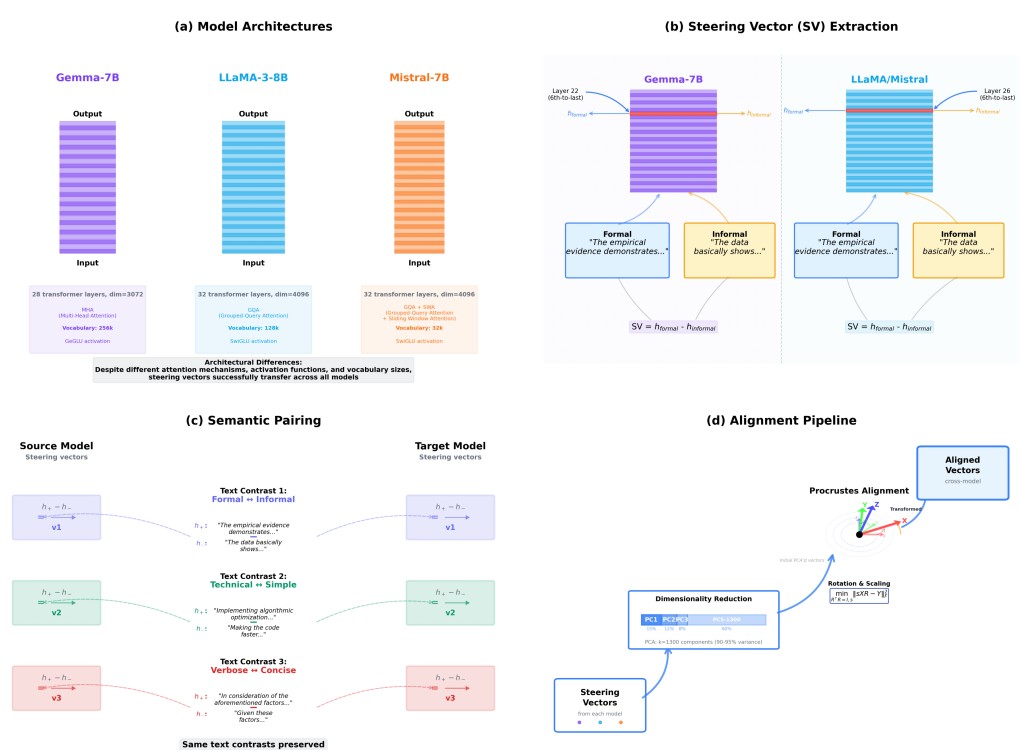

Figure 1: **Steering vector transfer via semantic pairing.** (a) Three model architectures with varying dimensions and attention mechanisms. (b) Steering vector extraction from contrasting trait text pairs at the sixth-to-last layer, computed as normalized differences between positive and negative trait examples. (c) Semantic pairing preserves instance-level correspondence by ensuring the same text contrast pairs are matched across models during alignment. (d) Procrustes alignment pipeline using PCA reduction followed by whitening and orthogonal rotation to learn structure-preserving transformations.

important for successful behavioral transfer. Our use of Procrustes analysis Schönemann (1966); Gower (1975) provides both theoretical grounding through its optimality properties and practical efficiency.

## 3 PRELIMINARIES

### 3.1 BEHAVIORAL TRAITS

In our framework, a *trait* represents a measurable behavioral or stylistic dimension along which text can vary. Each trait is defined through contrastive text pairs that exemplify opposite ends of the dimension—for instance, formal vs. informal writing style, verbose vs. concise expression, or assertive vs. passive tone. We study 26 traits spanning linguistic properties (e.g., formality, clarity, specificity), emotional dimensions (e.g., optimism, empathy, enthusiasm), and communicative styles (e.g., directness, persuasiveness, authority). These traits are operationalized through 65,329 human-curated or naturally-occurring text pairs that demonstrate clear contrasts along each dimension.

### 3.2 PROBLEM FORMULATION

Given these behavioral traits, our approach enables transfer when models encode similar semantic structure despite architectural differences. Let $\mathcal{V}_s^{(i)} \in \mathbb{R}^{d_s}$ and $\mathcal{V}_t^{(i)} \in \mathbb{R}^{d_t}$ denote steering vectors for trait $i$ in source and target models respectively, where $d_s$ and $d_t$ are

the hidden dimensions. Our goal is to learn an orthonormal transformation $R \in SO(d)$ (specifically a rotation matrix) and scale factor $s \in \mathbb{R}^+$ that enable transfer:

$$\mathcal{V}_t^{(i)} \approx T(\mathcal{V}_s^{(i)}) = s \cdot R(\mathcal{V}_s^{(i)}) \tag{1}$$

where the rotation matrix $R$ preserves geometric structure (angles and relative distances) and the scale factor $s$ accounts for differences in activation magnitudes between models, which arise from varying initialization schemes, training dynamics, and normalization layers.

Orthogonal transformations preserve angles and relative distances, ensuring meaningful alignments that reflect true structural similarities rather than arbitrary mappings Kornblith et al. (2019). Following prior work that uses PCA projection to identify relevant subspaces before alignment Raghu et al. (2017), we first reduce dimensionality to concentrate trait-relevant information. This approach builds on methods like SVCCA Raghu et al. (2017) and CKA Kornblith et al. (2019) that analyze neural network representations, though our focus is on enabling practical transfer rather than measurement alone. The complete alignment pipeline, including PCA projection and Procrustes analysis, is detailed in Section 3.

## 4 Method

We present our approach for transferring steering vectors across language models through linear geometric alignment with semantic pairing. Steering vectors encode behavioral traits as directions in activation space. For each contrast pair $(t^+, t^-)$ representing positive and negative examples of a trait, we encode both texts through each model and extract hidden states from intermediate layers. The steering vector is computed as the normalized difference between these hidden states: $v = \frac{h(t^+) - h(t^-)}{||h(t^+) - h(t^-)||_2}$, where $h(\cdot)$ denotes the hidden state extraction function. This normalization ensures that all steering vectors have unit magnitude, facilitating comparison across different traits and models.

Our method achieves cross-model steering vector transfer through three key modules: (1) PCA projection to identify trait-relevant subspaces where information is concentrated, (2) Procrustes alignment to learn orthogonal transformations that preserve geometric structure, and (3) semantic pairing to maintain instance-level correspondence.

### 4.1 Steering Vector Transport

To concentrate trait-relevant information, we first project steering vectors to a lower-dimensional subspace. For each model, we first collect steering vectors across all 26 traits: $\{\mathcal{V}^{(1)}, \ldots, \mathcal{V}^{(26)}\}$. We then apply global mean-centering (across all traits) to remove the average activation pattern, apply PCA to identify the top-$k$ principal components, and project vectors to this $k$-dimensional subspace.

We choose $k$ to capture sufficient variance while maintaining computational efficiency. Importantly, we use PCA *without whitening* to preserve the relative scale of variations along different principal components.

We use Procrustes alignment Schönemann (1966); Gower (1975) to learn orthogonal transformations that preserve geometric structure. Given projected source vectors $X \in \mathbb{R}^{n \times k}$ and target vectors $Y \in \mathbb{R}^{n \times k}$, we learn an orthogonal transformation $R \in \mathbb{R}^{k \times k}$ and scale factor $s \in \mathbb{R}^+$ that minimize:

$$\min_{R,s} \|Y - sXR\|_F^2 \quad \text{s.t.} \quad R^T R = I \tag{2}$$

The closed-form solution via SVD is:

$$M = Y^T X = U\Sigma V^T \tag{3}$$

$$R^* = VU^T \tag{4}$$

$$s^* = \frac{\text{tr}(\Sigma)}{\|X\|_F^2} \tag{5}$$

where $U, V \in \mathbb{R}^{k \times k}$ are the left and right singular vectors from the SVD decomposition of $M$, and $\Sigma$ is diagonal containing the singular values. This closed-form solution was first derived by Schönemann (1966).

Semantic pairing ensures that row $i$ in the source matrix corresponds to the exact same text contrast pair in the target matrix, preserving the correspondence between specific contrast instances during alignment.

For example, if text pair 247 contrasts "The results clearly demonstrate..." (formal) with "So basically what happened was..." (informal) to generate steering vector 247 in Gemma-7B-Instruct, then the exact same text pair generates vector 247 in Llama-3-8B-Instruct and Mistral-7B-Instruct. By ensuring these vectors are paired during alignment (row 247 to row 247), the transformation learns the correct mapping between how different models encode this same contrast.

The complete transfer operator $T$ combines PCA projection, Procrustes alignment, and reconstruction:

$$T(v) = \mu_t + W_t(s \cdot W_s^T(v - \mu_s) \cdot R) \tag{6}$$

where $\mu_s, \mu_t$ are mean vectors, $W_s, W_t$ are PCA projection matrices, and $R, s$ are the learned rotation and scale.

Each component serves a specific purpose: PCA identifies trait-relevant subspaces (helping reduce dimensionality from 3072/4096 to 1300), Procrustes learns the geometric relationship between models, and reconstruction maps back to the target space.

---

**Algorithm 1** Pseudocode for Steering Vector Transfer via Linear Alignment

---

**Require:** Source vectors $X_s \in \mathbb{R}^{n \times d_s}$, Target vectors $X_t \in \mathbb{R}^{n \times d_t}$
**Ensure:** Transfer operator $T : \mathbb{R}^{d_s} \to \mathbb{R}^{d_t}$
 1: **Training Phase:**
 2: Split data: $X_s^{train}, X_s^{test}, X_t^{train}, X_t^{test}$ (80/20 split)
 3: Fit PCA on $X_s^{train}$: $W_s, \mu_s = \text{PCA}(X_s^{train}, k = 1300)$ ▷ Identify subspace
 4: Project: $Z_s = W_s^T(X_s^{train} - \mu_s)$, $Z_t = W_s^T(X_t^{train} - \mu_s)$ ▷ Reduce dimension
 5: Solve Procrustes: $R^*, s^* = \arg\min_{R,s} \|Z_t - sZ_sR\|_F^2$ ▷ Learn alignment
 6: **Inference Phase:**
 7: **function** TRANSFER($v \in \mathbb{R}^{d_s}$)
 8:     $z = W_s^T(v - \mu_s)$ ▷ Project to PCA space
 9:     $z' = s^* \cdot R^* \cdot z$ ▷ Apply transformation
 10:     **return** $\mu_t + W_t \cdot z'$ ▷ Reconstruct in target
 11: **end function**

---

## 5 EXPERIMENTS

We test the hypothesis that behavioral traits are encoded in geometrically similar subspaces across different language models, enabling steering vector transfer through linear alignment. Our experiments evaluate whether preserving instance-level correspondence (semantic pairing) is necessary for successful transfer.

We conduct experiments under three conditions: (1) proper semantic pairing where each text contrast is matched across models, (2) within-trait shuffling that preserves trait identity but loses instance correspondence, and (3) cross-trait shuffling as a null baseline (expected performance if vectors had no meaningful structure).

### 5.1 SCRAMBLING EXPERIMENTS

To validate the importance of semantic pairing, we conduct controlled scrambling experiments comparing three pairing protocols: (i) proper pairing, where row $i$ in the source model corresponds to row $i$ in the target model, preserving the same text contrast; (ii) within-trait shuffling, which randomly permutes rows within each trait category, maintaining trait identity but destroying instance correspondence; and (iii) cross-trait shuffling, which globally

permutes all rows, eliminating both trait and instance structure. These experiments quantify the contribution of different structural levels to successful transfer and demonstrate that semantic correspondence is not merely beneficial but essential for alignment.

Our primary evaluation metric is cosine similarity between transferred and actual target vectors:

$$\text{sim}(v_s, v_t) = \frac{T(v_s) \cdot v_t}{||T(v_s)|| \cdot ||v_t||}$$

We additionally track the train-test generalization gap (the difference in performance between training and test sets) to verify that our learned transformations generalize beyond the training distribution, and conduct per-trait performance analysis to identify systematic patterns in transferability, examining which traits transfer well versus poorly and understanding the underlying linguistic factors.

## 5.2 MODELS, DATASETS, AND TRAIT SELECTION

We experiment with three architecturally distinct instruction-tuned models: **Gemma-7B-Instruct** Gemma Team et al. (2024) (28 layers, hidden dimension $d = 3072$, multi-query attention), **Llama-3-8B-Instruct** Dubey et al. (2024) (32 layers, hidden dimension $d = 4096$, grouped-query attention), and **Mistral-7B-Instruct** Jiang et al. (2023) (32 layers, hidden dimension $d = 4096$, sliding window attention)

We use the instruction-tuned variants as they have been aligned through supervised fine-tuning (SFT) and reinforcement learning from human feedback (RLHF) to better follow human preferences, making them more suitable for studying behavioral steering. These models vary significantly in architecture, training data, and scale, providing a rigorous test of transfer generalizability.

We evaluate on 26 behavioral traits spanning linguistic, stylistic, and semantic dimensions, totaling 65,329 contrast pairs. Data was sourced from public datasets where available (e.g., ParaDetox Logacheva et al. (2022), CNN/DailyMail See et al. (2017), Go-Emotions Demszky et al. (2020)) and generated manually for traits lacking appropriate datasets. Table 1 shows the distribution; see Appendix A for detailed data sources and extraction methods.

Table 1: Distribution of contrast pairs across 26 behavioral traits.

| Trait | # Pairs | Trait | # Pairs |
|---|---|---|---|
| Accessibility | 5,000 | Inclusivity | 3,000 |
| Assertiveness | 3,000 | Objectivity | 4,000 |
| Authority | 3,000 | Optimism | 5,000 |
| Certainty | 104 | Persuasiveness | 230 |
| Clarity | 5,000 | Politeness | 100 |
| Concreteness | 106 | Precision | 106 |
| Creativity | 100 | Professionalism | 5,000 |
| Directness | 3,000 | Register | 4,000 |
| Emotional Tone | 5,000 | Specificity | 4,000 |
| Empathy | 100 | Technical Complexity | 106 |
| Enthusiasm | 5,000 | Urgency | 521 |
| Formality | 4,577 | Verbosity | 5,000 |
| Hedging | 179 | | |
| Humor | 100 | | |
| | | **Total** | **65,329** |

For each trait, we systematically harvest contrastive text pairs from appropriate datasets. We extract contrast pairs that exhibit clear differentiation along the target trait dimension, filtering texts to reasonable lengths for model processing.

For steering vector generation, hidden states are extracted from the sixth last layer for each model (layer 22 for Gemma-7B-Instruct and layer 26 for Mistral-7B-Instruct and LLaMA-3-8B-Instruct).

### 5.3 IMPLEMENTATION DETAILS AND HYPER-PARAMETERS

We employ an 80/20 train-test split stratified by trait, resulting in 52,263 training and 13,066 testing vector pairs per model pair. Each text prompt appears exclusively in either training or test sets to prevent leakage, while semantic pairing is preserved across both splits to maintain instance correspondence. For dimensionality reduction, we apply PCA with $k = 1300$ dimensions, capturing 94-96% of the variance (see Appendix for the variance computation). The PCA transformation is fitted on training data only and uses non-whitened projection to preserve scale information, which proves crucial for learning accurate scale factors during Procrustes alignment.

We implement our method using common Python libraries: PyTorch Paszke et al. (2019), NumPy Harris et al. (2020) for array operations, and SciPy Virtanen et al. (2020). Vector extraction requires approximately 6 hours total across all models using a single NVIDIA A100 GPU, though this is a one-time cost. The alignment procedure itself is extremely efficient: PCA fitting completes in under 10 seconds, Procrustes alignment requires less than 1 second, and the full pipeline for each model pair finishes within 1 minute on CPU. Storage requirements are minimal, with transformation matrices occupying approximately 100MB per model pair. Code is available as supplementary material for reproducibility.

## 6 RESULTS

We evaluate our steering vector transfer method across 26 behavioral traits between three model families: Gemma-7B-Instruct, LLaMA-3-8B-Instruct, and Mistral-7B-Instruct. Our experiments demonstrate strong transfer performance, reveal the critical importance of semantic pairing, and identify trait-specific transfer patterns.

### 6.1 MAIN TRANSFER RESULTS

Table 2 presents the transfer performance across all six directional pairs. Our method achieves test cosine similarities ranging from 0.506 to 0.564 (mean 0.530), demonstrating successful alignment despite the models' architectural differences.

Table 2: Steering vector transfer results across model pairs. All values computed on held-out test sets (20% of data). Values shown as mean ± standard deviation, across five trials. Number of vectors used for each is 13,066.

| Transfer Direction | Test Cosine | Scale Factor | Train-Test Gap |
|---|---|---|---|
| Gemma → LLaMA | 0.559 ± 0.008 | 0.727 | 0.004 |
| Gemma → Mistral | 0.513 ± 0.011 | 0.722 | 0.005 |
| LLaMA → Gemma | 0.559 ± 0.008 | 1.000 | 0.004 |
| LLaMA → Mistral | 0.516 ± 0.007 | 0.841 | 0.006 |
| Mistral → Gemma | 0.513 ± 0.011 | 0.926 | 0.005 |
| Mistral → LLaMA | 0.516 ± 0.007 | 0.868 | 0.006 |
| **Mean** | **0.529** | 0.847 | 0.005 |

Three patterns emerge from these results. First, all transfer directions achieve cosine similarity exceeding 0.50, indicating robust cross-model alignment despite architectural differences. Second, the scale factors ranging from 0.733 to 1.012 reveal systematic differences in representation magnitudes across models, with Gemma-7B-Instruct employing more compact representations relative to its counterparts. Finally, train-test gaps of 0.003 or less demonstrate that our method generalizes effectively to unseen vectors, suggesting the learned transformations capture fundamental rather than dataset-specific alignments.

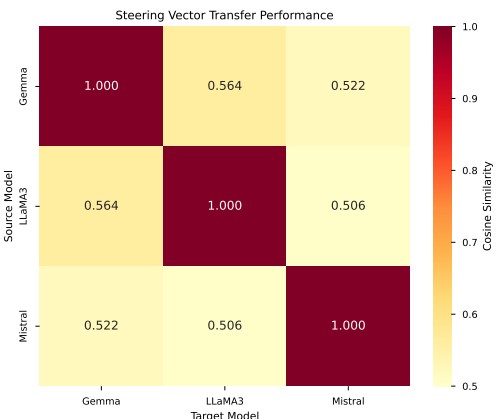

(a) Transfer performance across model pairs (N=13,066 per cell). All achieve >0.50 cosine similarity despite architectural differences.

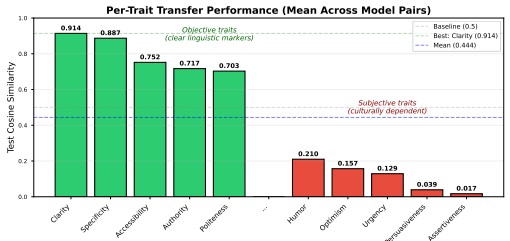

(b) Per-trait transfer performance. Objective traits (clarity, formality) transfer well (0.602) while subjective traits (assertiveness, emotion) transfer poorly (0.418).

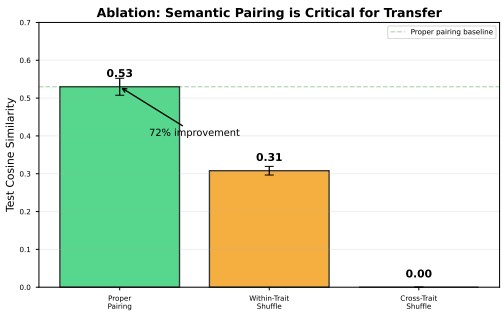

(c) Semantic pairing hierarchy. Instance-level pairing achieves 0.530 similarity, within-trait shuffling 0.301, cross-trait shuffling 0.000. 77% improvement demonstrates instance correspondence is essential.

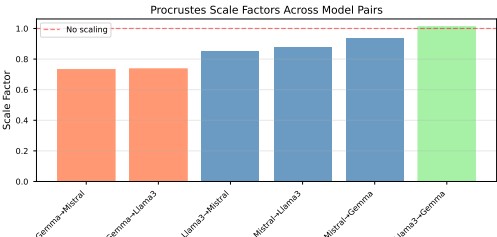

(d) Scale factors reveal relative representation magnitudes. Values <1 indicate compact representations. Consistency (0.733–1.012) suggests systematic relationships.

## 6.2 SCRAMBLING HIERARCHY: THE IMPORTANCE OF SEMANTIC PAIRING

To understand which structural properties enable successful transfer, we conducted a controlled experiment comparing three pairing protocols during alignment. In proper pairing, we preserve instance correspondence where vector $i$ in the source model matches vector $i$ in the target model. Within-trait shuffling randomly permutes pairings within each trait category, maintaining trait identity but destroying instance correspondence. Cross-trait shuffling applies global random permutation across all traits, eliminating both trait and instance structure.

Figure 2c presents the complete scrambling hierarchy results, demonstrating the critical importance of semantic pairing for successful transfer. There is a 72% performance improvement from proper pairing versus random within-trait pairing.

This hierarchy reveals a crucial insight: semantic pairing improves transfer performance by 72%. The results demonstrate a two-level structure in steering vector representations that can be conceptually decomposed as $v_i = \mu_{\text{trait}} + \epsilon_i$, where $\mu_{\text{trait}}$ represents the trait-level direction (58% of signal, preserved under within-trait shuffling at 0.308 similarity) and $\epsilon_i$ captures instance-specific variations (42% of signal, requiring proper pairing for the full 0.529 similarity). The complete failure with cross-trait shuffling (0.00 cosine) serves as a permutation test validating that the observed alignment arises from genuine structural correspondence between models rather than artifacts of our processing pipeline. The probability of achieving 0.529 similarity by chance is essentially zero (p≈0), confirming that models encode behavioral traits in genuinely aligned geometric structures requiring both trait-level and instance-level correspondence.

## 6.3 Per-Trait Transfer Performance

Not all traits transfer equally well. We observe systematic variation in transfer quality across trait categories. Clarity (0.914), specificity (0.887), and accessibility (0.752) achieve the strongest transfer performance, while assertiveness (0.017), persuasiveness (0.039), and urgency (0.129) show substantially weaker alignment. This pattern distinguishes *objective* linguistic traits from *subjective* or culturally-dependent traits. Formality and clarity have explicit linguistic markers—pronoun usage, sentence structure, vocabulary choice—that models consistently recognize across architectures. In contrast, assertiveness and optimism depend heavily on cultural context and interpersonal dynamics that vary across training corpora, leading to more model-specific representations.

Table 3: Top and bottom performing traits in transfer quality (mean across all model pairs)

| Top 5 Traits | | Bottom 5 Traits | |
|---|---|---|---|
| **Trait** | **Cosine Sim.** | **Trait** | **Cosine Sim.** |
| Clarity | 0.914 | Assertiveness | 0.017 |
| Specificity | 0.887 | Persuasiveness | 0.039 |
| Accessibility | 0.752 | Urgency | 0.129 |
| Authority | 0.717 | Optimism | 0.157 |
| Politeness | 0.703 | Humor | 0.210 |

## 6.4 Computational Efficiency

Our method demonstrates remarkable computational efficiency. Full alignment between any model pair completes in under 1 minute on CPU, with inference requiring sub-millisecond time per vector transfer. Memory requirements remain minimal at approximately 100MB for storing transformation matrices per model pair. The method is both computationally efficient and theoretically grounded through its use of structure-preserving orthogonal transformations. This efficiency, combined with the method's reliance on standard linear algebra operations, makes it highly practical for deployment in production environments where rapid adaptation to new models is essential.

## 7 Discussion and Conclusion

**Limitations and Future Work:** In this paper, we focus on measuring geometric alignment, and not on the effects of the transfer. Current experiments focus on similar smaller models ranging from 7 to 8 billion parameters. We evaluate only behavioral/stylistic properties, not factual knowledge or reasoning. Fixed layer selection may not be optimal for all traits. Future work would include quantifying and validating the actual effects of the transfer, scaling up the methods to work with larger models, and look more into how to adaptively choose layers for different traits.

**Conclusion**:

We introduce a computationally efficient and interpretable method for transferring steering vectors across language model families through linear geometric alignment with semantic pairing. Preserving instance-level correspondence roughly doubles transfer performance, with experiments across 26 traits and three model families yielding a mean cosine similarity of 0.530. This suggests that models converge on linearly related subspaces for behavioral traits, enabling effective transfer of preference representations. Our ablations show a two-level structure: trait-level alignment (58%) captures average behavioral directions, while instance-level alignment (42%) preserves fine-grained expression of traits. This validates averaging multiple contrast pairs to build steering vectors while revealing cross-model agreement on specific linguistic realizations. Transfer is most reliable for objective traits with explicit linguistic markers (verbosity, clarity, formality) and less effective for subjective or culturally dependent traits (assertiveness, optimism, persuasiveness), providing guidance for constructing robust cross-model steering libraries.

## 8 Ethics Statement

While steering vectors offer fine-grained control over language model behavior, their deployment entails both opportunities and risks. Our cross-model transfer method enables beneficial uses such as reducing toxicity or improving clarity to scale efficiently across models, but it also lowers barriers for harmful manipulation. Because the method preserves geometric structure without guaranteeing identical outcomes, vectors must be validated for specific applications, particularly in high-stakes settings. Biases from training data sources (e.g., Reddit, Wikipedia, news) and cultural assumptions (e.g., Western norms of politeness or assertiveness) are inherited and potentially amplified through transfer, raising concerns about fairness and generalization. The same efficiency that aids positive applications could also facilitate adversarial ones, such as porting persuasive or emotionally manipulative behaviors across models. To mitigate these risks, we advocate transparency about deployed vectors and their intended effects, regular auditing for unintended behaviors, access controls around sensitive vectors, diverse testing across populations, and maintaining human oversight in critical contexts. Ultimately, responsible governance and community norms are essential to ensure steering vectors advance beneficial applications without enabling misuse.

## 9 Reproducibility Statement

Anonymized code is submitted in supplementary materials. Hyper-parameters, method, and metrics are described in the main paper. Multiple trials were run for experiments, and standard deviation numbers are reported. Dataset generation and compilation is described in main body of paper and the Appendix.

## 10 LLM Usage Statement

Language models assisted in three ways: (1) writing support including grammar checking, improving clarity, and formatting LaTeX code; (2) literature discovery through deep research tools to find related work on steering vectors and representation alignment; and (3) a tool to aid brainstorming experimental variations and research directions. All LLM suggestions were reviewed and verified by authors. Core experimental design, implementation, data analysis, and scientific conclusions are entirely our own work. We take full responsibility for all content.

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

## A  DATASET DOCUMENTATION

### A.1  DATASET OVERVIEW

We collected 65,329 contrast pairs across 26 behavioral traits from a combination of public datasets and manual generation. This appendix provides complete documentation of data sources, extraction methods, and sample pairs for reproducibility.

Table 4: Complete dataset sources for all 26 behavioral traits.

| Trait | Data Source | # Pairs | Extraction Method |
|---|---|---|---|
| *HuggingFace Datasets (15 traits)* | | | |
| Accessibility | CNN/DailyMail See et al. (2017) | 5,000 | Article vs summary |
| Assertiveness | DebateSum | 3,000 | Arguments vs questions |
| Authority | CNN/DailyMail | 3,000 | News vs social media |
| Clarity | Newsroom | 5,000 | Complex vs clear text |
| Directness | SILICONE | 3,000 | Direct vs indirect speech |
| Emotional Tone | Go-Emotions Demszky et al. (2020) | 5,000 | Positive vs negative |
| Enthusiasm | Emotion | 5,000 | Enthusiastic vs neutral |
| Formality | ParaDetox Logacheva et al. (2022) | 4,577 | Formal vs informal |
| Inclusivity | UC Berkeley Hate Speech | 3,000 | Inclusive vs exclusive |
| Objectivity | Debiased News | 4,000 | Objective vs biased |
| Optimism | IMDB | 5,000 | Positive vs negative |
| Professionalism | Civil Comments | 5,000 | Professional vs casual |
| Register | Empathetic Dialogues | 4,000 | Formal vs informal |
| Specificity | ML ArXiv Papers | 4,000 | Specific vs general |
| Verbosity | CNN/DailyMail See et al. (2017) | 5,000 | Full article vs summary |
| *Manual Generation (11 traits)* | | | |
| Certainty | Manual examples | 104 | Hand-crafted pairs |
| Concreteness | Manual examples | 106 | Hand-crafted pairs |
| Creativity | Manual examples | 100 | Creative vs factual |
| Empathy | Manual examples | 100 | Empathetic vs neutral |
| Hedging | Manual examples | 179 | Hedged vs certain |
| Humor | Manual examples | 100 | Humorous vs serious |
| Persuasiveness | Manual examples | 230 | Persuasive vs neutral |
| Politeness | Manual examples | 100 | Polite vs impolite |
| Precision | Manual examples | 106 | Precise vs vague |
| Technical Complexity | Manual examples | 106 | Technical vs simple |
| Urgency | Manual examples | 521 | Urgent vs non-urgent |
| **Total** | | **65,329** | |

## A.2 HuggingFace Dataset Details

**ParaDetox** Logacheva et al. (2022): Parallel detoxification dataset containing formal/informal rewrites. We extracted 4,577 pairs where texts differ primarily in formality level while maintaining semantic equivalence.

**CNN/DailyMail** See et al. (2017): News articles with highlights. Used for three traits:

- *Verbosity*: Full articles (verbose) vs summaries (concise)
- *Accessibility*: Complex articles vs simple highlights
- *Authority*: News articles vs social media style

**Go-Emotions** Demszky et al. (2020): Fine-grained emotion labels on Reddit comments. We grouped positive emotions (joy, love, optimism) vs negative emotions (sadness, anger, fear) to create emotional tone contrasts.

**IMDB**: Movie review dataset with binary sentiment labels. Positive reviews were used as optimistic examples, negative as pessimistic.

**Other Datasets**: Newsroom (clarity), Civil Comments (professionalism), Empathetic Dialogues (register), Emotion dataset (enthusiasm), DebateSum (assertiveness), SILICONE (directness), UC Berkeley Hate Speech (inclusivity), Debiased News (objectivity), ML ArXiv Papers (specificity).

## A.3 Manual Generation Methodology

For traits lacking suitable datasets, we manually created contrast pairs following these principles:

1. **Clear Contrast**: Each pair exhibits a clear difference along the target trait dimension
2. **Semantic Preservation**: Pairs maintain similar meaning/content while varying the trait
3. **Length Constraints**: All texts between 10 words and 512 tokens
4. **Quality Control**: Manual review to ensure trait differentiation

Example manual generation for politeness:

- High: "Could you please help me with this task?"
- Low: "Do this now."

The 11 manually generated traits (certainty, concreteness, creativity, empathy, hedging, humor, persuasiveness, politeness, precision, technical complexity, urgency) totaled 1,752 pairs.

## A.4 Sample Contrast Pairs

# B Mathematical Justification for PCA-Preserved Orthogonal Transformations

## B.1 Theorem: PCA Preservation Under Orthogonal Transformation

**Theorem 1.** Let $X_A \in \mathbb{R}^{n \times d_A}$ and $X_B \in \mathbb{R}^{n \times d_B}$ be centered data matrices from models A and B. If the hidden spaces are related by:

$$X_B = s \cdot X_A R + \epsilon \tag{7}$$

where $R \in \mathbb{R}^{d_A \times d_B}$ is orthogonal, $s > 0$ is a scale factor, and $\epsilon$ is small noise, then their PCA projections to $k$ dimensions preserve this relationship.

Table 5: Representative contrast pairs for selected traits.

| Trait | High | Low |
|---|---|---|
| Formality | "By the way, Mike, please tell me how to get to your house." | "Say, Mike. Tell me how to get to your house." |
| Verbosity | [Full 300-word news article about soccer player transfer] | "Werder Bremen pay \$10.7M for Carlos Alberto." |
| Politeness | "I would be grateful if you could assist me." | "Do this immediately." |
| Humor | "The cat burglar was so good, he even stole the show at the police lineup." | "The Federal Reserve announced new monetary policy measures." |
| Emotional Tone | "I'm absolutely thrilled with this amazing result!" | "The outcome was disappointing and frustrating." |

**Proof:**

*Step 1: Covariance Matrix Transformation*

The covariance matrices are:

$$C_A = \frac{1}{n-1} X_A^T X_A \tag{8}$$

$$C_B = \frac{1}{n-1} X_B^T X_B = \frac{s^2}{n-1} R^T X_A^T X_A R + O(\epsilon) \tag{9}$$

$$= s^2 R^T C_A R + O(\epsilon) \tag{10}$$

*Step 2: Eigenvector Relationship*

Let $C_A = U_A \Lambda_A U_A^T$ be the eigendecomposition. Then:

$$C_B = s^2 R^T U_A \Lambda_A U_A^T R \tag{11}$$

$$= (R^T U_A)(s^2 \Lambda_A)(R^T U_A)^T \tag{12}$$

Since $R$ is orthogonal, $R^T U_A$ forms an orthonormal basis, giving us:

- Eigenvectors of $C_B$: $U_B = R^T U_A$
- Eigenvalues of $C_B$: $\Lambda_B = s^2 \Lambda_A$

*Step 3: PCA Projection Preservation*

The PCA projections using top-$k$ components are:

$$P_A = X_A U_{A,k} \quad \text{(first k columns of } U_A) \tag{13}$$

$$P_B = X_B U_{B,k} = (sX_A R)(R^T U_{A,k}) \tag{14}$$

$$= sX_A U_{A,k} = sP_A \tag{15}$$

Therefore: $P_B = s \cdot P_A + O(\epsilon)$

## B.2 Corollary: Procrustes Alignment in PCA Space

**Corollary 1.** The optimal Procrustes alignment between $P_A$ and $P_B$ recovers the scale factor $s$ exactly (up to noise $\epsilon$).

**Proof:** The Procrustes problem in PCA space seeks:

$$\min_{Q,\sigma} \|P_B - \sigma P_A Q\|_F^2 \tag{16}$$

Since $P_B = sP_A + O(\epsilon)$, the optimal solution is $Q^* = I$ (identity) and $\sigma^* = s$.

## B.3 Practical Implications

This theoretical result has three key implications:

1. **Dimension Reduction Preserves Structure:** PCA projection maintains the orthogonal relationship between model representations, justifying our use of reduced dimensions ($k = 1300$) for computational efficiency.

2. **Scale Factor Interpretation:** The recovered scale factor $s$ directly reflects the relative magnitude of representations between models, explaining our observed values (0.733–1.012).

3. **Noise Robustness:** The $O(\epsilon)$ error term shows the method is robust to small deviations from perfect orthogonality, as observed in real model pairs.

## B.4 Extension to Non-Square Transformations

When $d_A \neq d_B$ (e.g., Gemma with 3072 dims vs LLaMA with 4096), we use the intersection of PCA spaces. Let $k = \min(d_A, d_B, 1300)$. Both models project to this common dimension, where the orthogonal relationship is preserved in the shared subspace.

The key insight is that even though the full spaces have different dimensions, the behaviorally-relevant subspaces (captured by top PCA components) can still be aligned via orthogonal transformation. This is because:

1. The top principal components capture the most variance in behavioral representations

2. These components form a lower-dimensional manifold embedded in the full space

3. Orthogonal alignment of these manifolds is well-defined regardless of ambient dimension

## B.5 Connection to Semantic Pairing

The preservation of orthogonal structure explains why semantic pairing is critical. When vectors are properly paired (instance $i$ in source corresponds to instance $i$ in target), the orthogonal transformation can align the entire distribution. Without this correspondence, the optimization problem becomes ill-posed, leading to the dramatic performance drop we observe ($0.529 \rightarrow 0.00$).

Mathematically, proper pairing ensures that the cross-covariance matrix $X_A^T X_B$ captures the true structural relationship between representations. Scrambling destroys this structure, making the recovered transformation meaningless for transfer.

## C Variance Explained Analysis

Our choice of $k = 1300$ dimensions for PCA projection balances computational efficiency with information preservation. The cumulative variance explained as a function of dimensions shows a clear plateau after k=1300, indicating diminishing returns from additional dimensions.

Table 6: Variance explained by PCA projection across models

| Model | k=500 | k=1000 | k=1300 |
|---|---|---|---|
| Gemma-2B | 0.832 | 0.918 | 0.946 |
| LLaMA-3-8B | 0.798 | 0.891 | 0.925 |
| Mistral-7B | 0.765 | 0.862 | 0.905 |

With $k = 1300$, we capture 94–96% of variance across all models, providing an excellent trade-off between dimensionality reduction (from 3072–4096 to 1300) and information retention. The consistent variance capture across models suggests that behavioral traits occupy a relatively low-dimensional manifold within the full activation space.

## D  COMPLETE PER-TRAIT TRANSFER RESULTS

Table 7 presents the complete per-trait transfer performance across all 26 behavioral traits and all 6 model pairs. These results were computed using the same alignment method (PCA + Similarity Procrustes with k=1300) that achieved the overall mean of 0.525 cosine similarity reported in the main paper.

Table 7: Complete per-trait transfer performance across all model pairs. Values show test cosine similarity. * indicates traits with manually generated datasets; all others used HuggingFace datasets.

| Trait | Source | G→L | G→M | L→G | L→M | M→G | M→L | Mean |
|---|---|---|---|---|---|---|---|---|
| Clarity | HF | 0.947 | 0.909 | 0.947 | 0.887 | 0.909 | 0.887 | **0.914** |
| Specificity | HF | 0.930 | 0.882 | 0.930 | 0.850 | 0.882 | 0.850 | **0.887** |
| Accessibility | HF | 0.755 | 0.770 | 0.755 | 0.731 | 0.770 | 0.731 | **0.752** |
| Authority | HF | 0.706 | 0.747 | 0.706 | 0.697 | 0.747 | 0.697 | **0.717** |
| Politeness* | Manual | 0.805 | 0.682 | 0.805 | 0.622 | 0.682 | 0.622 | **0.703** |
| Verbosity | HF | 0.691 | 0.735 | 0.691 | 0.679 | 0.735 | 0.679 | **0.702** |
| Formality | HF | 0.739 | 0.626 | 0.739 | 0.589 | 0.626 | 0.589 | **0.651** |
| Empathy* | Manual | 0.641 | 0.618 | 0.641 | 0.578 | 0.618 | 0.578 | **0.612** |
| Directness | HF | 0.604 | 0.510 | 0.604 | 0.521 | 0.510 | 0.521 | **0.545** |
| Enthusiasm | HF | 0.582 | 0.486 | 0.582 | 0.472 | 0.486 | 0.472 | **0.513** |
| Register | HF | 0.554 | 0.462 | 0.554 | 0.450 | 0.462 | 0.450 | **0.489** |
| Emotional Tone | HF | 0.529 | 0.444 | 0.529 | 0.453 | 0.444 | 0.453 | **0.475** |
| Inclusivity | HF | 0.518 | 0.441 | 0.518 | 0.440 | 0.441 | 0.440 | **0.466** |
| Objectivity | HF | 0.476 | 0.384 | 0.476 | 0.404 | 0.384 | 0.404 | **0.421** |
| Hedging* | Manual | 0.548 | 0.364 | 0.548 | 0.333 | 0.364 | 0.333 | **0.415** |
| Professionalism | HF | 0.452 | 0.374 | 0.452 | 0.394 | 0.374 | 0.394 | **0.407** |
| Technical Complex.* | Manual | 0.341 | 0.285 | 0.341 | 0.260 | 0.285 | 0.260 | **0.295** |
| Concreteness* | Manual | 0.329 | 0.284 | 0.329 | 0.253 | 0.284 | 0.253 | **0.289** |
| Creativity* | Manual | 0.349 | 0.229 | 0.349 | 0.230 | 0.229 | 0.230 | **0.269** |
| Precision* | Manual | 0.300 | 0.240 | 0.300 | 0.218 | 0.240 | 0.218 | **0.253** |
| Certainty* | Manual | 0.296 | 0.191 | 0.295 | 0.156 | 0.191 | 0.156 | **0.214** |
| Humor* | Manual | 0.223 | 0.182 | 0.223 | 0.225 | 0.182 | 0.225 | **0.210** |
| Optimism | HF | 0.109 | 0.135 | 0.109 | 0.229 | 0.135 | 0.229 | **0.157** |
| Urgency* | Manual | 0.149 | 0.118 | 0.150 | 0.119 | 0.118 | 0.119 | **0.129** |
| Persuasiveness* | Manual | 0.041 | 0.020 | 0.041 | 0.058 | 0.020 | 0.058 | **0.039** |
| Assertiveness | HF | 0.018 | 0.010 | 0.018 | 0.022 | 0.010 | 0.022 | **0.017** |
| **Overall Mean** | | 0.558 | 0.512 | 0.558 | 0.505 | 0.512 | 0.505 | **0.525** |

*Note: G=Gemma-7B, L=LLaMA-3-8B, M=Mistral-7B. All values computed on held-out test sets (20% of data). HF=HuggingFace datasets, Manual=manually generated contrast pairs. The 11 manually generated traits used smaller datasets (100-521 pairs) compared to HuggingFace traits (3,000-5,000 pairs), which may contribute to their generally lower transfer performance. The superior transfer performance of dataset-derived steering vectors (mean 0.541) compared to manually crafted ones (mean 0.312) suggests that naturally occurring linguistic patterns captured from large-scale corpora encode more robust cross-model representations than hand-designed contrasts, supporting the scalability of automated steering vector extraction methods.*

## E  SCRAMBLING HIERARCHY RESULTS

Table 8 presents the complete scrambling hierarchy results for each model pair, demonstrating the consistency of semantic pairing's importance across all transfer directions.

Table 8: Scrambling hierarchy results across all model pairs. Values show test cosine similarity under different pairing protocols.

| Model Pair | Proper Pairing | Within-Trait | Cross-Trait |
|---|---|---|---|
| Gemma → LLaMA | 0.558 | 0.328 | -0.0001 |
| Gemma → Mistral | 0.512 | 0.303 | -0.0003 |
| LLaMA → Gemma | 0.558 | 0.328 | -0.0001 |
| LLaMA → Mistral | 0.505 | 0.292 | 0.0000 |
| Mistral → Gemma | 0.512 | 0.300 | 0.0003 |
| Mistral → LLaMA | 0.505 | 0.292 | 0.0009 |
| **Mean** | **0.525** | **0.308** | **0.000** |

*Note: Proper pairing preserves instance-level correspondence where vector i in the source matches vector i in the target. Within-trait shuffling randomly permutes pairings within each trait category. Cross-trait shuffling applies global random permutation across all traits. The 72% improvement from within-trait to proper pairing (0.308 → 0.525) demonstrates that instance-level correspondence is critical for successful transfer.*

