# OpenReview forum: "Steering Vector Transfer via Orthonormal Transformations and Semantic Pairing"
_ICLR.cc/2026/Conference — ICLR 2026 Conference Withdrawn Submission_

### Official Review · Reviewer_M6ra · 2025-10-28

**Soundness:** 2
**Presentation:** 2
**Contribution:** 2
**Rating:** 4
**Confidence:** 4

**Summary:**

This paper studies the cross-model transferability of steering vectors, which is achieved using the proposed orthonormal transformations. To ensure an accurate transformation, the authors employ the semantic pairing (i.e.,  row i in the source matrix corresponds to the exact same text contrast pair in the target matrix). The authors experimentally show that: 1) cross-model transferability is possible, 2) semantic pairing enhances transferability, and 3) objective linguistic traits transfer better than subjective ones.

**Strengths:**

1. A orthonormal-transformation-based method for cross-model transferability of steering vectors is proposed
2. The authors show that objective linguistic traits transfer better than subjective ones.

**Weaknesses:**

The paper's primary contribution appears to be significantly diminished by an incomplete review of prior studies. A body of existing literature has already explored the transferability of steering vectors, including both cross-lingual [1] and cross-model scenarios [2].

Specifically, previous research[2] has established that simple linear transformation can effectively enable behavioral control and steering vector alignment across diverse LLMs. Such transformations can handle various traits and even lead to emergent weak-to-strong generalization. Consequently, the method proposed, which imposes an orthogonality constraint on linear transformations, should be framed as an incremental refinement rather than a very novel method.

In this light, the paper's contributions are more limited than claimed. The finding that objective linguistic traits exhibit superior transferability is an interesting observation, but it is derived exclusively under the orthonormal assumption, and its broader applicability remains unverified. Most critically, the concept of "semantic pairing" is presented as a significant insight. However, maintaining correspondence between input and output variables is a standard and essential precondition for any constrained optimization or supervised learning task. For example, it is axiomatic that one must preserve the integrity of data-label pairs (e.g., (x, y)) during model training. Therefore, highlighting the performance drop from "shuffling labels" (i.e., Not using semantic pairing)—a methodologically unsound practice—serves more as a strawman comparison than as a validation of a novel principle.

Reference
1. https://aclanthology.org/2024.findings-emnlp.96.pdf
2. https://aclanthology.org/2025.acl-long.185.pdf

**Questions:**

See weakness

---

### Official Review · Reviewer_JrJq · 2025-11-01

**Soundness:** 2
**Presentation:** 2
**Contribution:** 2
**Rating:** 2
**Confidence:** 4

**Summary:**

This paper studies the cross-model transfer of steering vectors between different LLM families. The method aligns paired texts across models and learns an orthonormal map to transport steering directions. While the paper introduces a compelling approach to alignment, it focuses primarily on the construction of the mapping without adequately validating how effective the alignment is in practice.

**Strengths:**

- The paper addresses an interesting problem of transferring steering vectors across models, which not only has practical implications but also provides evidence toward a universal geometry in LLM representations.
- Evaluation is conducted on a broad dataset covering 26 traits, demonstrating the method’s versatility.
- The proposed method is simple and easy to implement, which enhances its reproducibility and accessibility.

**Weaknesses:**

- Prior work on transferring or aligning representations and steering vectors is not sufficiently discussed (e.g., [1]). At minimum, the paper should include a comparative summary and a results table against representative baselines such as orthogonal Procrustes, unconstrained linear maps, CCA/PLS, and causal-tracing/latent-alignment variants.
- Although the claims span Gemma, LLaMA, and Mistral model families, experiments are only conducted on Gemma-7B-Instruct, LLaMA-3-8B-Instruct, and Mistral-7B-Instruct. To better support claims of generality, at least one larger model (e.g., 13B/30B/70B), one smaller model (e.g., 2B/3B), and one base (non-instruction-tuned) variant should be included to assess whether instruction tuning contributes significantly to alignment.
- The current evaluation focuses on geometric fit, but does not directly assess the downstream effectiveness of transferred steering. The paper should include an analysis of performance degradation (if any) between original and transferred steering vectors. Without behavioral validation, it remains unclear whether the alignment is functionally meaningful, or whether any alignment exists at all. The method relies on a strong assumption that hidden state geometry is similar across models, which is not empirically validated.
- The paper states that the transformation in Eq. 1 is rotational and scaled, but can $R$ also represent a reflection? What would be the implications if this were the case?
- Section 5.2 selects the sixth-to-last layer across all models (lines 324–326) without any justification. Clarification or ablation over different layers would strengthen the choice.
- Text in Figure 1 is too small and difficult to read. The visualization should be made clearer and more self-contained; currently, it depends too much on the main text for interpretation.
- References to the appendix are not hyperlinked, which hinders navigation.
- How was class imbalance handled during training and evaluation? Could PCA be biased toward dominant traits due to data imbalance?
- Please report the memory complexity of the full pipeline, including hidden state extraction and PCA computation.
- A cosine similarity around 0.5 may not be intuitively meaningful. Can the authors explain how to interpret this magnitude in context? What constitutes a “good” similarity score in this setting?

**Questions:**

See Weaknesses.

---

### Official Review · Reviewer_sgkg · 2025-11-01

**Soundness:** 2
**Presentation:** 2
**Contribution:** 1
**Rating:** 2
**Confidence:** 3

**Summary:**

They transfer steering vectors across models from different families with different architectures. In particular, they first PCA the steering vectors to get a set of principal components, then learn the transfer in PC space, then reconstruct the final steering vector. They also claim that semantic pairing is important for transfer performance. They use the cosine similarity between the transferred vector and the steering vector if it was done on the new model, and find that they can get a cosine similarity of up to 0.56.

**Strengths:**

* Doing steering vector transfer through Principal Components first is an interesting idea.
* They demonstrate their idea across several model families.

**Weaknesses:**

* Steering vector transfer has been done before (e.g. by [Huang et al. (2025)](https://arxiv.org/abs/2501.02009v2)) and is not particularly novel.
* The only metric used in this paper to measure transfer is cosine similarity. What does a cosine similarity of 0.5 mean? Is it twice as good as 0.25? Ultimately, since steering vectors are used to influence model behavior, it is insufficient to only use cosine similarity to measure the effectiveness of transfer. You also need to look at steering outcomes.
* There is also a lack of comparable baselines in terms of methods. For example, semantic pairing might be only relevant to this proposed method where it is sensitive to individual prompts.

**Questions:**

Clarifications
* How is this work meaningfully different from Huang et al. (2025)?
* The steering vector here is calculated as differences between hidden states of a contrast pair (Line 186). Can I clarify that this means that $n$ in line 205 is the number of total contrast pairs then, and not the number of traits?
* Follow-up Question: If so, doesn't this mean that the learned transformation is sensitive to the prompt pairs (the $n$ dimension), which naturally explains why semantic pairing is important? With a more typical method like differences-in-mean, where you first take the mean vector across the set of prompts, the semantic pairing finding would not apply.

Feedback
* The Figure fonts are far too small to be read.
* Most of the citations should be done in parentheses with `citep{}`.
* The font is different from the standard ICLR submission, which makes it somewhat harder to read.
* L176: ...is detailed in Section 3 > Should this be 4?

---

### Official Review · Reviewer_kBxn · 2025-11-01

**Soundness:** 2
**Presentation:** 2
**Contribution:** 2
**Rating:** 2
**Confidence:** 4

**Summary:**

The authors demonstrate the transfer of steering vectors of behavioral traits between the models using a method based on dimensionality reduction and procrustes alignment. They compare three modes: 1) sentence-pairing, 2) trait-level pairing, and 3) trait scrambling and report the cosine similarity of the transferred vector to the target vector. The cosine similarity is the highest at the sentence level matching, compared to trait pairing only, and the scrambling gives a near-zero. They compare the performance per trait, showing that objective traits are better transferred than the subjective ones.

**Strengths:**

1. Well designed experiments: good baseline comparison on multiple levels: trait matching and full scrambling.

2. Empirical difference is convincing: A Clear relative difference in cosine similarities between the comparing modes.

3. Extensive dataset : The Authors curated and additionally collected a dataset for diverse behavioral traits.

**Weaknesses:**

1. Cosine similarity as the only metric.
I am mostly concerned that a mere geometric measure of transferred vector (cosine similarity) might be not significant for actual ‘behavior transfer’ which is the ultimate goal.. I agree that there is a clear empirical difference in the cosine similarity from the baseline, but the authors do not verify the actual behavioral difference elicited by this steering vector. It is plausible that the high cosine similarity won’t guarantee the intended behavioural shift. Direct evidence on the behavioral change by injecting the transferred vector is needed to support what the authors claim and show the validity of the proposed approach. In a similar vein, there is a comparative difference in cosine similarity between different methods, but this might not be reflected ultimately in the generated outputs.

2. Train / test split
The authors use a train/test split that covers the same behavioral traits. With the semantic pairing, I think finding high cosine similarity is somewhat obvious in the current split regime. To claim representational universality, especially from the Platonic Representation Hypothesis view, I think a fairer setting would be having a disjoint train/test set of traits. I think the current method and experimental results show that identical stimuli yield partially correlated representations, rather than representational universality. And this seems to be already reflected in the result of objective behavior being much easier to transfer, since there are probably higher lexical/template similarities than true ‘traits’.

**Questions:**

Here are minor questions and comments.

1. How sensitive is the result to PCA ‘k’ selection?
2. Why the 6th to last layers? How sensitive is the result to the choice of the layers for hidden states extraction?
3. There are lots of citations that miss parentheses within the sentence. E.g. line 40
4. It was confusing for me to understand if the steering vector is collected per sentence and used for alignment, or use the averaged/common direction of those as a single trait vector. Also, some notions are not clearly defined (e.g. line 205  - I had to infer what is dimension n )
5.It seems like there is a mismatch in the numbers in Figure 2c and in the text. In the figure, it is 53% and often in the text, it reports 58%.


Given the insufficient evidence, I’m giving 2 - I’m willing to increase my score if the authors improve/clarify the points in the weaknesses.

**Details Of Ethics Concerns:**

Ethics concern was cleary mentioned in the Ethics Stament.

---

### Note · Authors · 2025-12-03

**Comment:**

We're deeply grateful to the reviewers for their time and effort. In particular, we believe the emphasis on the verification of steering outputs' effectiveness and necessary comparison to (Huang et al. 2025) are important additions to the work and require time and effort to be done carefully. We are working on a rewrite of the paper to address these concerns appropriately. We believe this will require major revisions to the work---as a result, we have decided to withdraw the current submission to make these changes. We are grateful for the feedback, which will improve the final version of the paper substantially.

**Withdrawal Confirmation:**

I have read and agree with the venue's withdrawal policy on behalf of myself and my co-authors.